# An Investigation of the Operating Principles and Power Consumption of Digital-Based Analog Amplifiers

Anna Richelli *,†, Paolo Faustini †, Andrea Rosa †and Luigi Colalongo †

Department of Information Engineering, University of Brescia, Via Branze 38, 25123 Brescia, Italy;
p.faustini001@studenti.unibs.it (P.F.); andrea.rosa@unibs.it (A.R.); luigi.colalongo@unibs.it (L.C.)
* Correspondence: anna.richelli@unibs.it; Tel.: +39-030-371-5501
† These authors contributed equally to this work.

**Abstract:** Digital-based differential amplifiers (DDA) are particularly suitable to low voltage digital integrated circuit technologies. This paper presents an exhaustive analysis of digital-based analog amplifiers to take advantage of today's high-performance digital technologies, and of computer aided design (CAD), which is commonly employed to design integrated circuits. The operating principle and the main mathematical relations of digital-based differential amplifiers are discussed along with an exhaustive explanation of its operating regions and of the corresponding power consumption. These aspects, which are not discussed in the literature, are very important for the circuit designers. Finally, a detailed description of the design procedure of the UMC 180nm standard CMOS technology is provided.

**Keywords:** digital-based amplifiers; ultra-low power; CMOS integrated circuits





## 1. Introduction

Ubiquitous electronics require compact ultra-low power devices and fast prototyping. Technology scaling favors digital circuits thanks to their high speed and low power dissipation. In this context, there is an increasing trend of implementing low-voltage inverter-based analog circuits. An elementary inverter-based amplifier comprises a pair of CMOS digital inverters that operate in an analog fashion due to the common-mode (CM) voltage that keeps both the NMOS and PMOS transistors in the saturation region. The resulting amplifiers have high differential gain, large transconductance, high output resistance, and high gain bandwidth (GBW). The gain of the simple inverter is the ratio of the small signal transconductances to output conductances: $A_v = (g_{mN} + g_{mP})/(g_{dsN} + g_{dsP})$, where $g_m$ is the transconductance and $g_{ds}$ the output conductance. Unfortunately, operating the inverter as an analog amplifier leads to a large variation in the DC gain and GBW due to the temperature and the fabrication process. Furthermore, the higher the voltage gain is, the narrower is the range of the CM becomes. To address these issues, several techniques have been proposed in the literature, mostly based on common mode feedback (CMFB) circuits, which sense the output CM to control the bias current of the inverter. Indeed, $g_{mN}$, $g_{mP}$, and $g_{dsN}$, $g_{dsP}$ depend on the drain and source voltages and, in turn, on the CM. Therefore, the DC gain can be regulated by changing the CM. In principle, only two resistors are required to extract the CM. The CMFB can also be implemented without resistors, as in the Nauta operational transconductance amplifier and its most recent derivations [1–5]. Nevertheless, they remain analog circuits in which both the NMOS and PMOS are in the saturation region, the current flows continuously, and the level of static power consumption is high. Several approaches, based on current-starved topologies, have been suggested in the literature to reduce power consumption; such approaches involve the use of fully differential circuits that need an output common mode voltage feedback to stabilize the small signal performances.

Another approach is to radically rethink analog functions in digital terms [6,7], using only digital circuits, such as a class D amplifier [8]. VCO-based amplifiers [9–11], voltage-to-time converters [12,13], charge amplifiers [14], analog-to-digital [15–19] and digital-to-analog converters [20], digital voltage references [21], low-dropout regulators (LDO) [22], hybrid analog-digital amplifiers [23] and digital-based amplifiers [24–32] have been reported in the literature. In [24–32], a differential amplifier, composed of only logic gates, was proposed. It has several appealing features, such as low power consumption, small area, easy design, and fast prototyping. It is an interesting approach, and a deep understanding of the possible topologies, designs, features, and limits is important for analog-background designers, who habitually use different design methodologies. This paper is focused on the understanding of the DDA from both a circuital and mathematical standpoints, with particular emphasis on its power consumption, which is one of its main appealing features. In Section 2, the operating principles, the transistors operating conditions, and the main mathematical relations required to design the DDA are devised. In Section 3, the full design of the amplifier in 180 nm CMOS standard is shown, along with a comprehensive explanation of the operating regions and power consumption. In Section 4, some conclusions are drawn.

## 2. Operating Principle of the DDA

The building blocks of the DDA are shown in Figure 1 [24]. The output of the digital buffers ($OUT+$, $OUT-$) is high (H) when the input voltages $v_{i+}$, $v_{i-}$ are larger than the threshold voltage ($V_M$) and low (L) when are lower than $V_M$. The CMFB (green box) adjusts the CM voltage $v_{CM}$ in order to emulate the input stage of a differential amplifier. Hence, $OUT+$ and $OUT-$ are related to the differential voltage $v_D = v_{i+} - v_{i-}$. A detailed description of the DDA, along with the basic mathematical relations, are reported in [24]. For the sake of clarity, here we recall: $v_{CM} = (v_{i+} + v_{i-})/2$, $v_{i\pm} = v_{CM} \pm v_D/2$ and, using a balanced resistor network, $v'_{i\pm} = (v_{i\pm} + v_{CM})/2$.

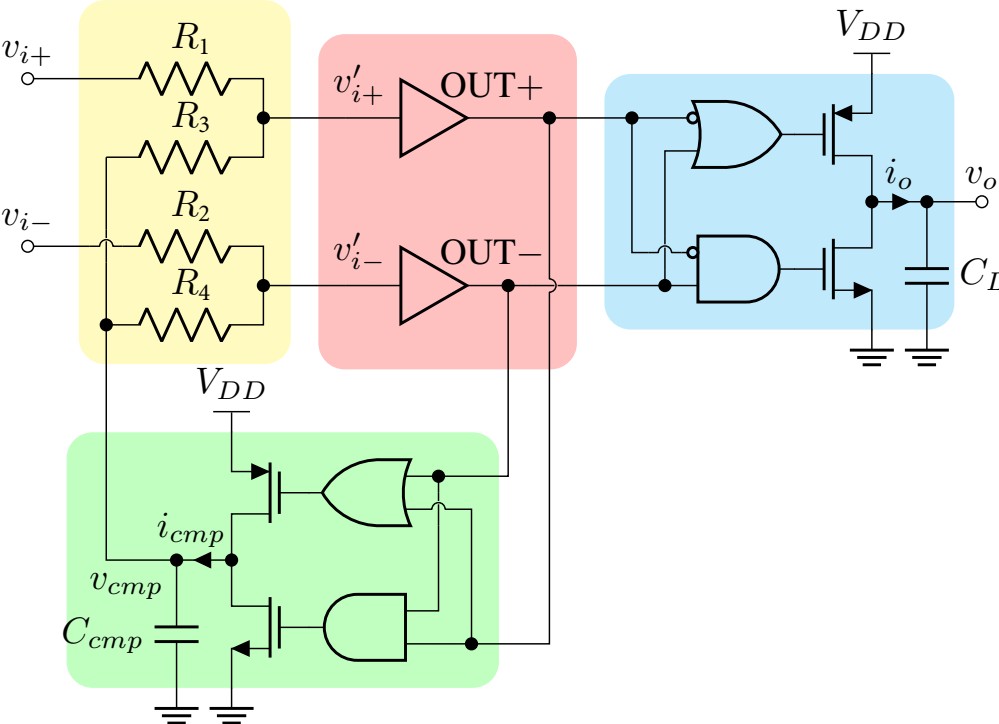

**Figure 1.** Digital-based amplifier [24], resistor network (yellow box), digital buffers (pink box), output stage (blue box), CMFB (green box). $R_1 = R_2 = R_3 = R_4 = R$.

Since the logic gates are assumed to switch much faster than the input signals of the DDA, the output and CMFB voltages $v_o$ e $v_{cmp}$ are well approximated by the following first order differential equations:

$$\frac{dv_o(t)}{dt} = \frac{i_o(t)}{C_L}, \quad \frac{dv_{cmp}(t)}{dt} = \frac{i_{cmp}(t)}{C_{cmp}} \tag{1}$$

where $i_o(t) = \{ i_o^{pmos}$ if $[v'_{i+}(t_1) > V_M] \wedge [v'_{i-}(t_1) < V_M], -i_o^{nmos}$ if $[v'_{i+}(t_1) < V_M] \wedge [v'_{i-}(t_1) > V_M], 0$ elsewhere $\}$, $i_{cmp} = \{ i_{cmp}^{pmos}$ if $[v'_{i+}(t_2) < V_M] \wedge [v'_{i-}(t_2) < V_M], -i_{cmp}^{nmos}$ if $[v'_{i+}(t_2) > V_M] \wedge [v'_{i-}(t_2) > V_M], 0$ elsewhere $\}$, $t_1 = t - t_{D,o}$, $t_2 = t - t_{D,cmp}$ and $t_{D,o}$ and $t_{D,cmp}$ represent the propagation time through the logic gates. Furthermore, for the sake of simplicity, the logic gates are assumed ideal with the same propagation time $t_D = t_{D,o} = t_{D,cmp}$, and $i_o^{pmos} = i_o^{nmos} = I_o$, $i_{cmp}^{pmos} = i_{cmp}^{nmos} = I_{cmp}$ are constant and not dependent on $v_o$ and $v_{cmp}$. Under those assumptions, in the following, the two possible operating conditions $v_D = 0$ and $v_D \neq 0$, will be discussed.

The waveforms are shown in Figure 2 in the case of $v_D = 0$. When $v_D = 0$, $OUT+$, $OUT-$ are the same, the output inverter is in high impedance, $v_o$ is constant, and the load capacitor $C_L$ holds its charge. Nevertheless, the compensation voltage $v_{cmp}$ oscillates. In fact:

- At $t = 0$, both $v'_{i+}$ and $v'_{i-}$ cross $V_M$, and $OUT+$, $OUT-$ switch from L to H. It takes a certain amount of time $t_D$ for the signal to propagate through the CMFB;
- Before $t_D$, although $OUT+$, $OUT-$ are high, the CM compensation inverter has not yet changed its state, and $C_{cmp}$ is still charging with $I_{cmp}$ as when $t = 0$;
- At $t = t_D$, the CMFB changes its state: the pull-down switches on, and the capacitor $C_{cmp}$ is discharged with a constant current $-I_{cmp}$;
- From $t_D$ to $2t_D$, while $C_{cmp}$ is discharging, both $v'_{i+}$ and $v'_{i-}$ fall below the threshold $V_M$;
- At $2t_D$, $OUT+$, $OUT-$ switch from H to L;
- Before $3t_D$, the CM compensation inverter has not yet changed its state and $C_{cmp}$ is still discharging;
- At $3t_D$, the CM compensation inverter changes state, the pull-up switches on, and the capacitor $C_{cmp}$ is charged with the current $I_{cmp}$;
- This cycle is repeated every $T_{cmp} = 4t_D$.

Hence, the delay introduced by the compensation network $t_D$ induces a triangular wave oscillation on $v_{cmp}$ of period $4t_D$ and peak-to-peak amplitude $v_{cmp,pp} = 2t_D I_{cmp}/C_{cmp}$.

In Figure 3, the waveforms when $v_D > 0$ are shown, similar considerations hold when $v_D < 0$:

- The differential voltage $v_D$ corresponds to a small mismatch between $v'_{i+}$ and $v'_{i-}$ that, in turn, causes $v'_{i-}$ to cross the threshold voltage $V_M$ with a small delay $\Delta t_C$; during $\Delta t_C$, the differential voltage is positive, $v'_{i+} > v'_{i-}$, and the outputs $(OUT+, OUT-) = (1, 0)$. After $2t_D$, it is $v'_{i+}$, that crosses the threshold voltage $V_M$ with a small delay $\Delta t_C$ respect to $v'_{i-}$.
- $v_{cmp}$ is a triangular wave with the same period $T_{cmp} = 4t_D$, as in Figure 2 but, during the interval $\Delta t_C$, the voltage is clamped since the buffer is in the high impedance region.
- During the interval $\Delta t_C$, the output buffer charges $C_L$ and $v_o$ steps up of $I_o \Delta t_C/C_L$.
- The charge on $C_L$ is incremented by $I_o \Delta t_C$, twice every $T_{cmp} = 4t_D$.

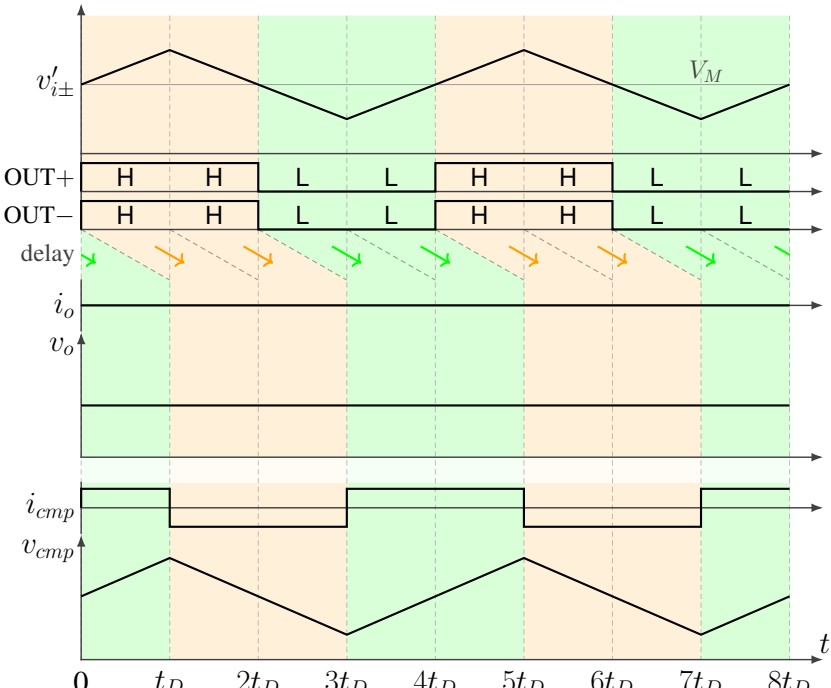

**Figure 2.** Waveforms of the DDA at $v_D = 0$, i.e., $v'_{i+} = v'_{i-}$.

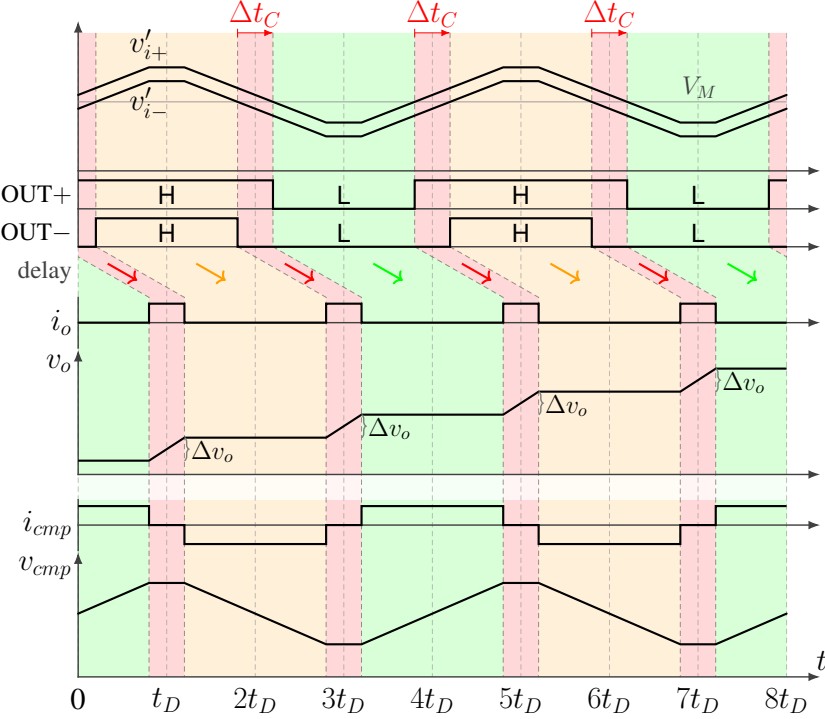

**Figure 3.** Waveforms of the DDA at $v_D > 0$, i.e., $v'_{i+} > v'_{i-}$.

In other words, the DDA operates a double conversion from voltage to time and back from time to voltage again. The first conversion is $v_D$ to $\Delta t_C$, thanks to the oscillation on $v_{cmp}$. Indeed, the mismatch on $v'_{i+}$ and $v'_{i-}$ is converted into a delay $\Delta t_C$, i.e., a time. Then, the output buffer converts the delay $\Delta t_C$ back into a voltage $\Delta v_o = I_o \Delta t_C / C_L$. The voltage gain of the DDA in the frequency domain, assuming only the capacitive load $C_L$, reads:

$$A_D(f) = \frac{G_D(f)}{j2\pi f C_L} = \frac{\alpha}{j2\pi f \, 2t_D} \, e^{-j2\pi f t_D} \tag{2}$$

where $\alpha = I_o / C_L$,

$$G_D(f) = \frac{I_o(f)}{V_D(f)} \approx \frac{I_o}{2t_D} \frac{C_{cmp}}{I_{cmp}} e^{-j2\pi f t_D} \tag{3}$$

and $e^{-j2\pi f t_D}$ is the phase shift due to the propagation delay $t_D$. Furthermore, when $f \ll f_c/2$, Equation (2) can be simplified as $A_D(f) \approx \alpha/(j2\pi f \, 2t_D)$. Thus, the transfer function is equivalent to an integrator with a unity gain frequency of $f_u = \alpha/(4\pi t_D)$. The digital-based analog amplifier can be used, almost as conventional analog amplifiers, in feedback loops. Nevertheless, the classical assumptions of infinite input impedance ($Z_i \to \infty$) and negligible output impedance ($Z_o \to 0$) are not properly verified. The transfer function can be approximated as:

$$G(f) = \frac{V_o(f)}{V_i(f)} = \frac{1/\beta}{1 + j2\pi f/(\beta f_u)} \tag{4}$$

where $\beta$ is the gain of the feedback network set by the resistors' ratio. In other words, the DDA operates as a first order system.

## 3. Design and Simulations of the DDA

The digital-based amplifier discussed in the previous section has been designed in the standard 180 nm United Microelectronic Corporation (UMC) CMOS process and extensively simulated in different operating conditions. It is worth noting that the final schematic is slightly modified with respect to the base circuit of Figure 1 to equalize the propagation time of $OUT+$ and $OUT-$. The circuit is extremely simple, composed only of resistors and logic gates. The supply voltage is standard for this technology (1.8 V), with the aim of investigating the DDA in normal operating conditions. Several simulations worked out at lower supply voltages show that the DDA operates correctly at a supply as low as 400 mV. The capacitive load is assumed of 10 pF. It is a fair value to account for typical operating conditions of the DDA, i.e., the parasitic effects of the pad, the bonding, the package, or the subcircuits connected as load. A buffer should be included if the DDA is connected to a bulky load.

### 3.1. Sizing of the DDA in UMC 180 nm CMOS Process

The first step is to dimension the CMOS inverter with a logic threshold $V_M$ half of $V_{DD}$. Considering the different transistor's threshold $V_{Tn,Tp}$ and mobility $\mu_{n,p}$, the ratio of the PMOS and NMOS, to maximize the symmetry of the inverter, is 4.725. Minimum transistors are chosen to minimize the gate capacitance: $L_{n,p} = 180$ nm, $W_n = 240$ nm, and $W_p = 1134$ nm. The NOR logic gate is composed of two PMOS connected in series and two NMOS in parallel. To have a symmetric behavior both the transistors of the pull-up and pull-down read $L_{n,p} = 180$ nm, $W_n = 240$ nm, and $W_p = 2268$ nm. Dual considerations hold for the NAND that comprises two PMOS in parallel and two NMOS in series: $L_{n,p} = 180$ nm, $W_n = 480$ nm, and $W_p = 1134$ nm. The input resistors $R_1 - R_4$, on the one hand, should be large, in order to have a high input impedance; on the other hand, they should not be too large, in order to limit the area consumption and the thermal noise. Although same resistances of 145 kΩ were chosen to have an input differential impedance of 580 kΩ, in the layout $R_1 - R_4$, they were slightly modified to have true rail-to-rail common mode: $R_{1,2} = 140$ kΩ and $R_{3,4} = 150$ kΩ. The output and the CMFB are dimensioned to balance the propagation delay between the logic gates. The dimensions of the output stage are $L_{n,p} = 180$ nm, $W_n = 5$ μm and $W_p = 10$ μm. And for the inverter of the CMFB is: $L_{n,p} = 180$ nm, $W_n = 240$ nm and $W_p = 720$ nm. All the dimensions are in agreement to the technology minimum grid. It is worth noting that process variation and mismatch can affect the circuit behavior, and in particular the input offset voltage. It is indeed related to the mismatch of the logic threshold $V_M$ and to the asymmetry of the propagation delay through the logic gates. For example, when $v_D = v_{i+} - v_{i-} = 0$, if

$\Delta V_M > 0$, OUT$-$ will cross the logic threshold before OUT$+$, the resulting time difference is $\Delta t_C < 0$, given by

$$\Delta t_C = -\frac{C_{cmp}}{I_{cmp}} \cdot 2\Delta V_M \tag{5}$$

and the input offset voltage reads:

$$V_{OS,1} = -2\Delta V_M \tag{6}$$

A similar result can be also drawn for $\Delta V_M < 0$. Moreover, the input offset voltage can be affected also by a mismatch in the propagation delay $\Delta t_D$:

$$V_{OS,2} = -\frac{1}{2} \cdot \frac{I_{cmp}}{C_{cmp}} \cdot \Delta t_D \tag{7}$$

The overall input offset result is:

$$V_{OS} = V_{OS,1} + V_{OS,2} = -2\Delta V_M - \frac{1}{2} \cdot \frac{I_{cmp}}{C_{cmp}} \cdot \Delta t_D \tag{8}$$

### 3.2. Simulation Results

The DDA can hardly be simulated by means of the classical small-signal AC analysis tools, since the core of its operating principle is digital and is related to the oscillation of $v_{cmp}$. Time-expensive transient analyses are required to design and characterize the amplifier. The DDA has been simulated in two different configurations: open loop as a comparator, and closed loop as a feedback amplifier.

### 3.2.1. Open Loop

The amplifier is operated as a comparator. In the simulations, to generate a modulated differential voltage $v_D$ that emphasizes all the different operating regions of the amplifier, we used two input sinusoidal signals $v_{i+}$ and $v_{i-}$ of amplitude $V_{DD}$ with different frequencies: $v_{i+}$ 10 kHz and $v_{i-}$ 30 kHz. The transient simulation time is 100 µs. The waveforms of the most relevant voltages of the internal nodes are shown in Figure 4; the bottom x-axis represents the simulation time, the top x-axis the operating regions, the y-axis is the voltage at the nodes. In Figure 5, one can see that the DDA has five distinct behaviors that we call operating regions 1–5:

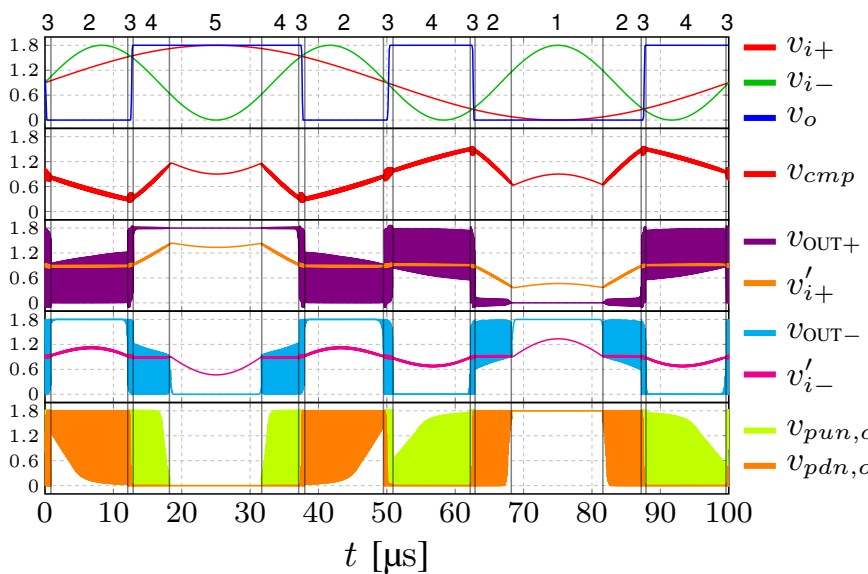

**Figure 4.** Open loop: input/output waveforms and intermediate node voltages.

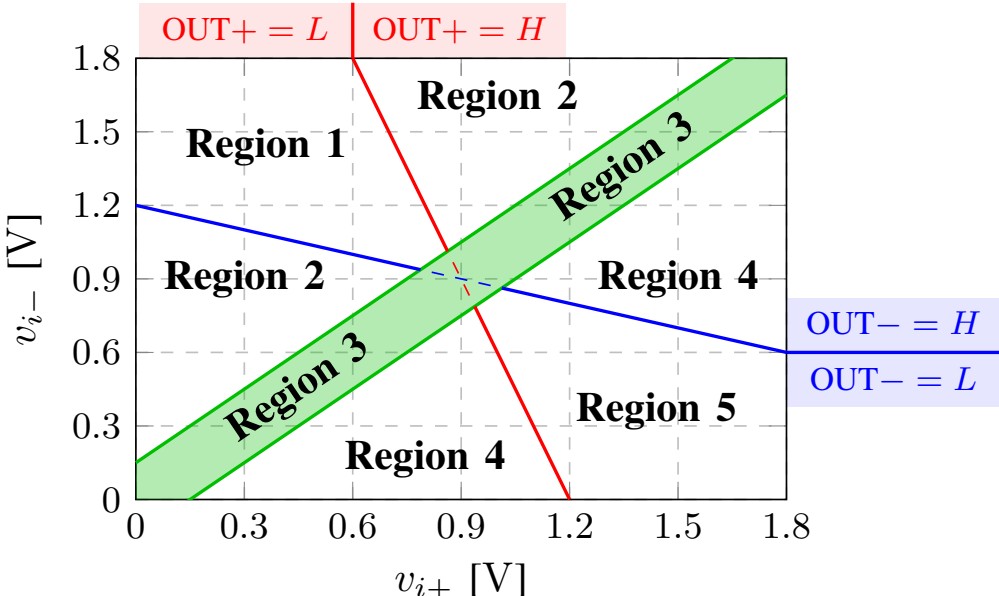

**Figure 5.** Open loop: operating regions vs. $v_{i\pm}$.

- In regions 1 and 5, the differential voltage $v_D$ is large, and $v_{i\pm}$ are well separated and opposite with respect to the logic threshold $V_M$. In these regions, the output voltage saturates to $V_{DD}$ or 0, the common-mode compensation network is not active, and only the pull-up or the pull-down of the output inverter turns on. In Figure 4, $v_{pun,o}$, $v_{pdn,o}$ are the gate voltages of the pull-up and pull-down, respectively. In Figure 6, regions 1 and 5 are limited by the equations $|v'_{i+}| < V_M$ and $|v'_{i-}| > V_M$:

$$v'_{i+} = \frac{v_{i+} R_3 + v_{cmp} R_1}{R_1 + R_3} \tag{9}$$

$$v'_{i-} = \frac{v_{i-} R_4 + v_{cmp} R_2}{R_2 + R_4} \tag{10}$$

Furthermore, since the CMFB is not active, $v_{cmp} = v_{CM}$, i.e., $(v_{i+} + v_{i-})/2$, regions 1 and 5 read:

$$v'_{i+} = \frac{v_{i+} + v_{CM}}{2} = \frac{3}{4} v_{i+} + \frac{1}{4} v_{i-} \tag{11}$$

$$v'_{i-} = \frac{v_{i-} + v_{CM}}{2} = \frac{1}{4} v_{i+} + \frac{3}{4} v_{i-} \tag{12}$$

- In region 3, the differential voltage $v_D$ is small enough to activate the CMFB. The compensation voltage $v_{cmp}$ oscillates and the digital outputs $OUT+$ and $OUT-$ commute between L and H. Both the pull-up and the pull-down of the output inverter are active; if $v_D$ is positive, $v_o$ steps up, if $v_D$ is negative, $v_o$ steps down. This region is defined by the condition $\Delta t_C < t_D$, i.e., $v_D < I_{cmp}/C_{cmp} t_D$.
- In regions 2 and 4, the differential voltage $v_D$ is small, but not as small as in region 3. In region 2, $v_D < 0$ and $OUT+$ holds the low logic state, while $OUT-$ quickly commutes from H to L due to the CMFB. The pull-down of the output stage switches on. In region 4, $v_D > 0$ and $OUT-$ holds the low logic state, while $OUT+$ quickly commutes from H to L due to the CMFB. The pull-up of the output stage switches on. Hence, the pull-up or the pull-down switches on, but are not always active as in regions 1 and 5.

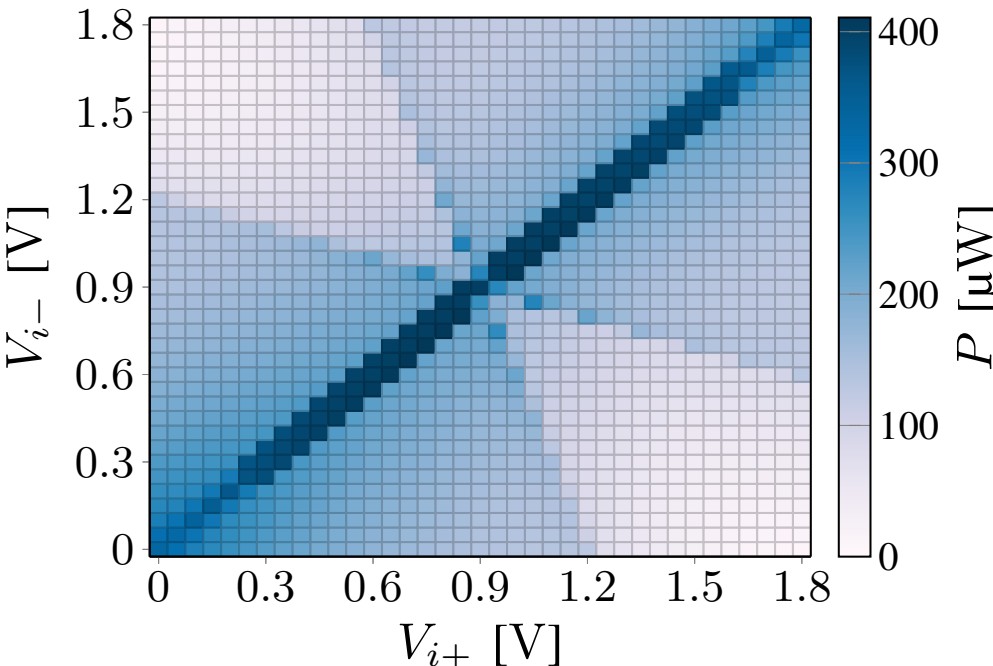

**Figure 6.** Open loop: power consumption vs. $v_{i\pm}$.

The power consumption of the DDA is mostly dynamic, and is due to the switching of the gates ($P_{gates}$) to the charging and discharging of $C_{cmp}$ ($P_{cmp}$) and $C_L$ ($P_o$). It strongly depends on the operating regions of the amplifier, as shown in Figure 6: the dissipated power, as a function of the differential voltage, is represented as a shade of blue from light (lower power consumption) to dark (higher power consumption). The x- and the y-axes are the input voltages $v_{i\pm}$, ranging from 0 to $V_{DD}$ in steps of 50 mV. It is worth adding that the simulations are worked out at 1.8 V (standard for this technology): if the voltage supply is reduced, the power consumption becomes remarkably smaller [26–29]. This is due both to the dependence of the dynamic power on $V_{DD}$ and to the reduction in the switching frequencies of the CMFB and of the output stage. In regions 1 and 5, the power dissipation is lower than in region 2, 3, and 4, since the common mode compensation network is always switched off, only the pull-up or pull-down is conducting, and the output voltage saturates to $V_{DD}$ or 0. In regions 2 and 4, the power dissipation is higher since the CMFB is active. Finally, in region 3, the power consumption reaches its maximum, since the differential voltage is small and $v_{cmp}$ oscillates continuously.

### 3.2.2. Closed Loop

The DDA can be used in feedback connection as an analog amplifier. In the simulations of the closed loop connection, a sinusoidal rail-to-rail input signal of 20 kHz is applied to the non-inverting input $v_{i+}$. The simulation time is 100 µs. The Fast Fourier Transform (FFT) of the buffer connection with unitary loop gain (G) is 0.992, the phase delay ($\varphi$) 0.08°, and the total harmonic distortion (THD) is 0.23%. Figure 7 shows that when the DDA is used as a buffer, the differential voltage $v_D$ is very small, and always operates in region 3. Hence, despite the rather good overall performances, the power consumption reaches its maximum.

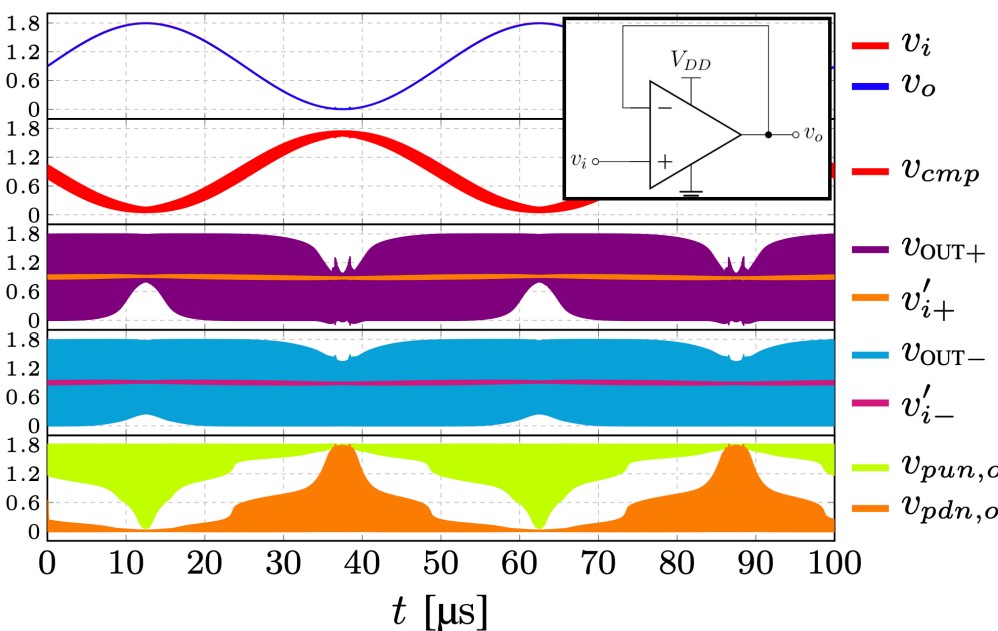

**Figure 7.** Buffer: input/output waveforms and internal node voltages.

Furthermore, the closed loop configuration has been simulated in several other configurations. Figure 8 shows the simulations in three different configurations: buffer (G = 0.987, $\varphi$ = 0.14°, THD = 0.31%), inverter (G = −0.938, $\varphi$ = 0.06°, THD = 1.58%), and gain two (G = 2.007, $\varphi$ = 0.06°, THD = 0.99%). The results are shown in Figure 8. Simulations are worked out with an input signal of 400 mVp and a frequency of 20 kHz. The voltage gain G and the output voltage $v_o$ are close to the ideal ones. While the largest THD is smaller than 1.6%, the phase delay is below 0.14°, and the larger offset 33.7 mV only.

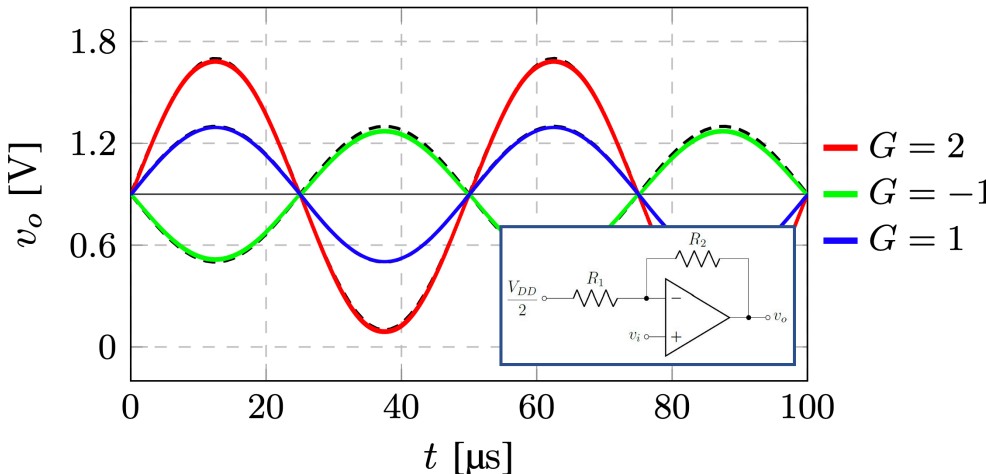

**Figure 8.** Closed loop simulations.

Finally, the closed loop amplifier is simulated with rail-to-rail input voltages at several frequencies. In Figure 9A, the FFT in the range of 100 kHz–100 MHz is shown.

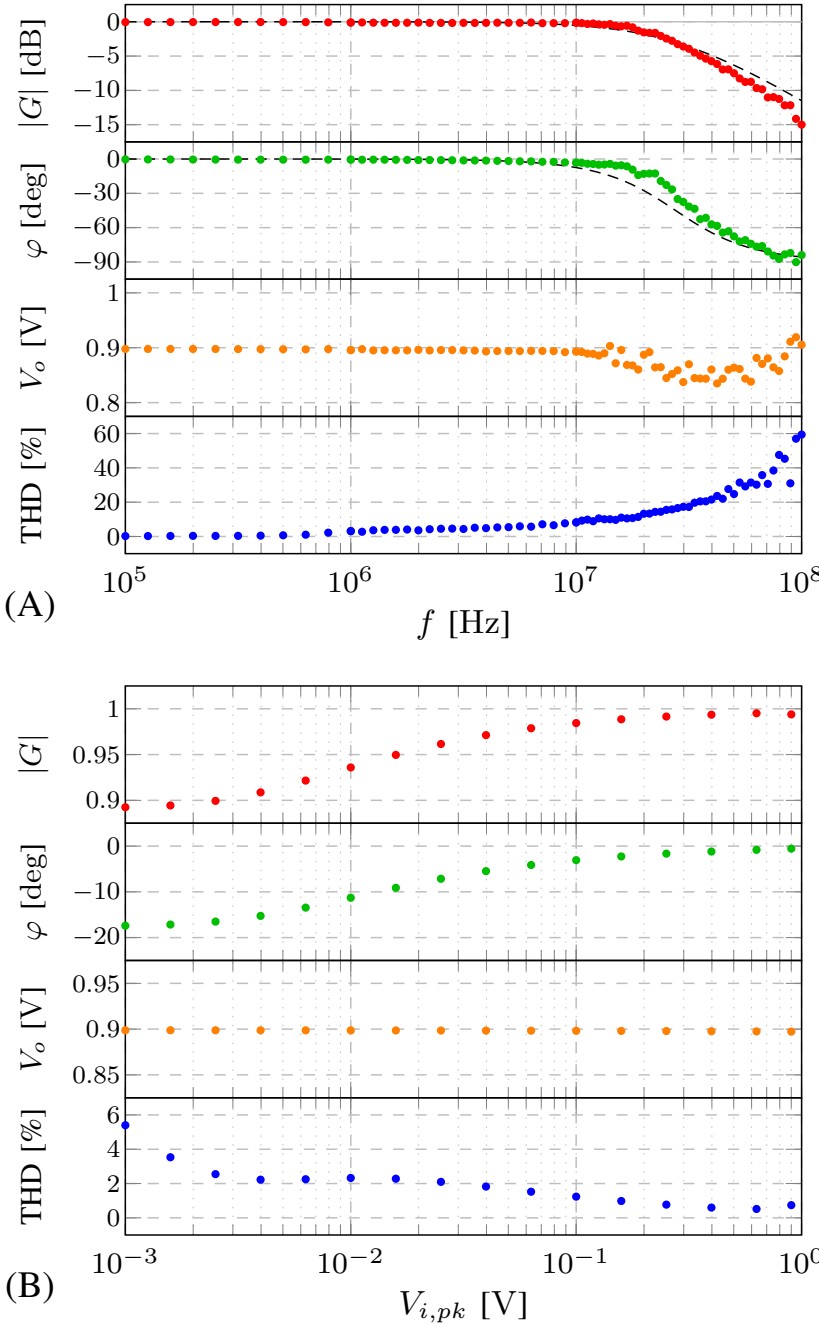

**Figure 9.** Corners of the buffer connection: G, $\varphi$, output voltage, and THD as a function of frequency (**A**), and of the input signal amplitude (**B**).

The amplifier acts as a single-pole dominant system with a unity gain frequency of about 27.6 MHz. The output voltage $v_o$ (gain and phase) is almost ideal up to 10 MHz, while the THD starts rising at a lower frequency of about 1 MHz. It is worth stressing that simulations are worked out at 1.8 V. Nevertheless, one of the most appealing features of the DDA is that the small power consumption and lower $V_{DD}$ are often used, and in that case, the overall performance degrades rather quickly. Finally, simulations were worked out by changing the amplitude of the input voltage at constant frequency (500 kHz), i.e., within the bandwidth of the amplifier. The results are shown in Figure 9B, for an input signal ranging from 1 mV up to 900 mV. Following the recent literature [32], the simulations were performed at different temperatures ranging from 27° to 40°. The corresponding results are shown in Figure 10: the simulations show that the circuit behavior is only slightly dependent on the temperature, in this range.

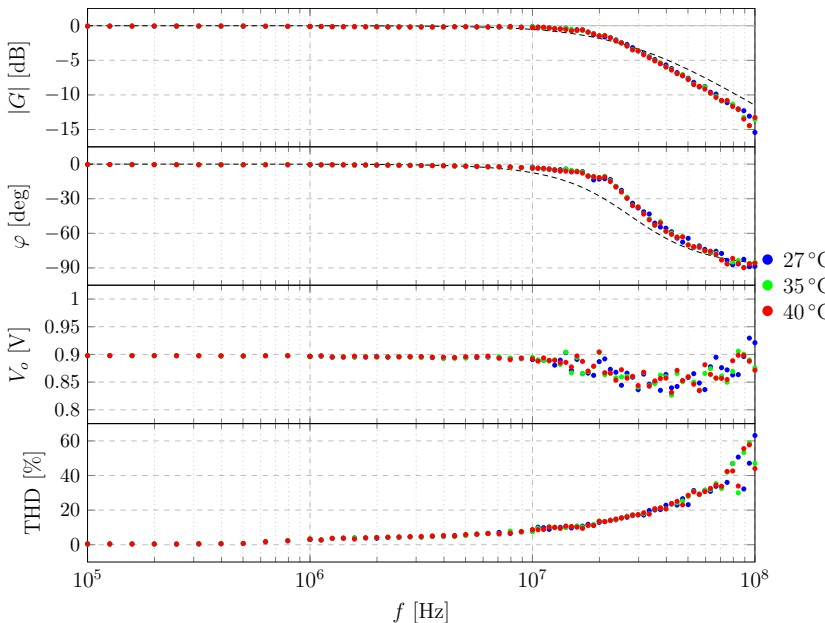

**Figure 10.** Corners of the buffer connection: G, $\varphi$, output voltage, and THD as a function of frequency, varying the temperature between 27° and 40°.

Furthermore, in order to deeply investigate the amplifier behavior, corner simulations were performed as well; the results are shown in Figure 11.

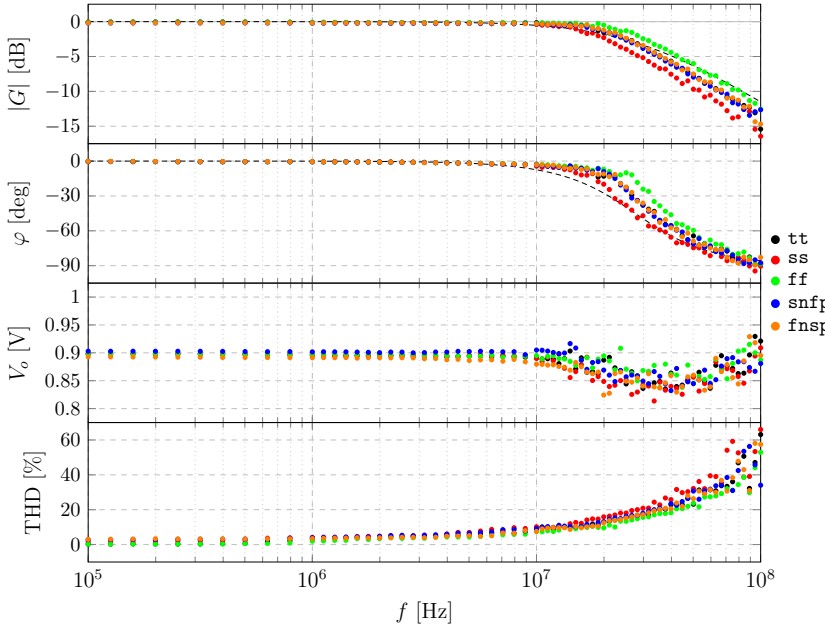

**Figure 11.** Corners of the buffer connection: G, $\varphi$, output voltage, and THD as a function of frequency.

As expected, the bandwidth of the amplifier is larger in the case of a fast–fast (FF) corner and narrower in the case of a slow–slow (SS) corner. The bandwidth does not change significantly in the case of SNFP and FNSP corners, since it depends on the average output current, that, in turn, depends on both the PMOS and the NMOS of the output stage (the last inverter in the blue box of Figure 1). In more detail: below 10 MHz, the gain and the phase are only barely dependent on the corners; above 10 MHz, the amplifier has a larger bandwidth in corner FF and narrower in corner SS; below 10 MHz, the gain and the phase are only barely dependent on the corners; and above 10 MHz, the amplifier has a larger bandwidth in corner FF and narrower in corner SS. The average value of the output voltage

Vo is only slightly dependent on the corners. Dealing with the THD, the corner SS is the worst case, while the corner FF is the best case since the THD depends on the transient response of the common mode feedback network (the green box of the Figure 1). A faster response leads to a better behavior of the amplifier, while a slower response deteriorates the THD. Indeed, the oscillation frequency of the CMFB signal (i.e., when the input voltages are both $v_{i+} = v_{i-} = 0.9$ V) peaks at the FF corner (302 MHz). In the other cases, it ranges between 216 MHz (corner SS), 260 MHz (corner SNFP), and 263 MHz (corner TT and FNSP). The corresponding power consumption is P = 400 µW in the TT corner; P = 270 µW in SS; P = 560 µW in FF; P = 384 µW in SNFP; and P = 410 µW in FNSP. Therefore, the corners SS and FF affect the oscillation frequency of the CMFB with a moderate impact on the power consumption and the THD. One can see that the DDA works very well when the amplitude of the input signal is large enough, and slightly deteriorates as input voltage becomes lower and lower. The resistive compensation network, in fact, limits the input impedance of the amplifier. In several applications, such as biomedical, wearable, IoT, and sensing, these limits do not represent a real drawback, and the DDA is a very promising architecture. High frequencies or low input signals represent the most important limit of the DDA that require some adjustments. To this aim, a more recent DDA architecture [29,32], with a new compensation-network-based on floating inverters, was reported. Thanks to this new common mode compensation circuit, the DDA in [29,32] can amplify low amplitude signals with very good overall performances. In order to highlight the characteristics of the DDA, a comparison with a sample of different types of amplifiers is shown in Table 1, where the digital-based amplifier is compared to the inverter-based amplifier and to other classical topologies (i.e., based on the standard differential pair, like the gate-driven amplifier, or the bulk-driven circuit).

**Table 1.** Comparison between DDA, Inverter-based, Bulk-driven and Gate-driven amplifiers.

| Architecture | DDA [28] | Inv-Based [33] | Bulk-Driven [34] | Gate-Driven [35] |
|---|---|---|---|---|
| Biasing | dynamic | static | static | static |
| Tecn. | 180 nm | 130 nm | 130 nm | 180 nm |
| Area | 1.4 µm$^2$ | - | 83 µm$^2$ | 0.08 mm$^2$ |
| $V_{DD}$ | 0.3 V | 0.3 V | 0.25 V | 0.4 V |
| DC gain | 31 dB | 49.8 dB | 60 dB | 60 dB |
| $C_L$ | 80 pF | 2 pF | 15 pF | 1 pF |
| GBW | 0.23 kHz | 9 kHz | 1.88 kHz | 1.2 MHz |
| Power | 0.4 nW | 1.8 nW | 18 nW | 10 µW |

The main advantages of the DDA are the ultra-low power consumption due to the dynamic biasing, the small area and the ease of design. The inverter-based amplifiers exhibit larger bandwidth, but the power consumption is higher. The advantages of the classical topologies are the gain and the bandwidth. On the other hand, they are difficult to design and they consume a lot of power.

## 4. Conclusions

A digital-based analog amplifier has been designed and investigated along with its main mathematical relations. We have shown that the amplifier can operate in five different regions, and that the power consumption peaks when $v_D$ is small. Several simulations have been worked out both in the open and closed loop configurations. The simulations show that the amplifier represents a really attractive approach for the signal conditioning of the integrated circuits with advantages on power consumption and ease of design. The amplifier is better suited for low- to medium-frequency input signals with rather large

amplitudes. Nevertheless, it represents a very appealing architecture when the area on chip and the power dissipation are of paramount importance.

**Author Contributions:** All the authors have contributed substantially to the paper. A.R. (Anna Richelli) and L.C. have supervised the work, have provided the simulation tools and have written the paper; P.F. is graduated-five years Laurea degree- student and A.R. (Andrea Rosa) is PhD student, they have investigated on the power consumption of the amplifier and performed the simulations. All authors have read and agreed to the published version of the manuscript.

**Funding:** This research received no external funding.

**Data Availability Statement:** Data not available in public archive.

**Conflicts of Interest:** The authors declare no conflict of interest.

### Abbreviations

The following abbreviations are used in this manuscript:

| | |
|---|---|
| DDA | Digital-based Differential Amplifier |
| CAD | Computer-aided Design |
| UMC | United Microelectronics Corporation |
| CMOS | Complementary Metal Oxide Semiconductor |
| CM | Common Mode |
| GBW | Gain Bandwidth |
| CMFB | Common Mode Feedback |
| DC | Direct Coupling |
| VCO | Voltage Controlled Oscillator |
| FFT | Fast Fourier Transform |
| G | Loop Gain |
| THD | Total Harmonic Distortion |
| IoT | Internet of Things |

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
