# Peer review of "An Investigation of the Operating Principles and Power Consumption of Digital-Based Analog Amplifiers"

_jlpea, doi:10.3390/jlpea13030051_

Round 1
Reviewer 1 Report
This paper presents the design of a digital-based analog amplifier (DDA) which architecture has been already presented in a paper of P. Crovetti en 2013. In the paper of Crovetti, the DDA was tested on a version using discrete components.
Here the DDA has been designed in a 180nm CMOS technology and simulation results are presented. The paper is interesting but doesn't provide sufficient results to be accepted for publication, especially according to the title which claim an investigation on the power consumption of the DDA. Only 13 lines discuss the power consumption (from line 134 to line 146). In addition the unit of power is not mentionned in figure 5 (is it µW ?).
So the paper needs to be improved by providing more simulation results discussing :
1) the effect of mismatches on the DDA performances, and especially on consumption. During the design, the authors claim that they have symmetrized the design as well as the propagation delay between the logic gates. I assume that any mismatch may have a strong influence on the DDA performance, especially on offset, as well as on consumption ? So Monte Carlo simulations need to be performed, and results commented
2) The temperature may also have a strong effect on consumption. Idem, simulations have to êrformed and the results commented concerning temperature effect.
3) The dimensions of the output stage are given but the output capacitance for which this output stage has been implemented is not provided... Please provide it, and comment on the output load effect.
Author Response
The authors thank the reviewers and the editor for the useful comments that help to improve the paper. We have carefully considered all the reviewer’s comments and we did our best to address, we hope, all his concerns. In the revised paper, all changes are marked in yellow.
Reviewer #1:
This paper presents the design of a digital-based analog amplifier (DDA) which architecture has been already presented in a paper of P. Crovetti en 2013. In the paper of Crovetti, the DDA was tested on a version using discrete components. Here the DDA has been designed in a 180nm CMOS technology and simulation results are presented. The paper is interesting but doesn't provide sufficient results to be accepted for publication, especially according to the title which claim an investigation on the power consumption of the DDA. Only 13 lines discuss the power consumption (from line 134 to line 146). In addition the unit of power is not mentionned in figure 5 (is it µW ?).
We have modified the title of the paper to be more correlated to the contents of the paper.
We have modified Fig 5 and introduced unit of power, and improved the overall quality of the figures.
So the paper needs to be improved by providing more simulation results discussing :
1) the effect of mismatches on the DDA performances, and especially on consumption. During the design, the authors claim that they have symmetrized the design as well as the propagation delay between the logic gates. I assume that any mismatch may have a strong influence on the DDA performance, especially on offset, as well as on consumption ? So Monte Carlo simulations need to be performed, and results commented
Unfortunately, Montecarlo simulations are not available on our design tool; nevertheless, following the reviewer comments, we have performed several simulations using corner models of pmos and nmos. Analogue designers usually want to know the effect of mismatches, so Montecarlo simulations are mandatory. On the other hand, the DDA, although emulates an analog amplifier, is a digital circuit and corner analysis considers the most extreme variations expected in process. With this information, it is possible to determine whether the circuit performance specifications will be met, when the random process variations combine in the most unfavorable patterns.
In the DDA the device mismatch affects the threshold voltage VM and the propagation delay with a subsequent increase of the offset; these considerations were included in the paper at the end of section 3.1.
Furthermore, we have simulated the open and closed loop amplifier behavior using corner models. As expected, the bandwidth of the amplifier is larger in the case of fast-fast (FF) and smaller in the case of slow-slow (SS) corner. The bandwidth does not change significantly in the case of SNFP and FNSP corners, because it depends on the average output current which, in turn, depends on both PMOS and NMOS transistors of the output stage (the last inverter in the blue box of the Fig.1). More in detail: below 10MHz, the gain and the phase are only barely dependent on the corners, above 10MHz, the amplifier has a larger bandwidth in corner FF and narrower in corner SS.
Besides, the average output voltage Vo is only slightly dependent on the corners.
Dealing with the THD, the corner SS is the worst case, while the corner FF is the best case since the THD depends on the transient response of the common mode feedback network (the green box of the Fig.1). A faster response leads to a better behavior of the amplifier, while a slower response slightly deteriorates the THD. Indeed, the oscillation frequency of the CMFB signal (when the input voltages are both vi+ = vi- = 0.9V) peaks at the FF corner (302MHz). In the other cases it ranges between 216MHz (corner SS), 260MHz (corner SNFP), 263MHz (corner TT and FNSP).
The corresponding power consumption is:
- TT: f = 263MHz, P = 400uW;
- SS: f = 216MHz, P = 270uW;
- FF: f = 302MHz, P = 560uW;
- SNFP: f = 260MHz, P = 384uW;
- FNSP: f = 263MHz, P = 410uW.
Therefore, the corners SS and FF affect the oscillation frequency of the CMFB output signal Vcmp, with a moderate impact on the power consumption and THD.
All the above considerations were included in the revised paper in the Section 3.2.
2) The temperature may also have a strong effect on consumption. Idem, simulations have to performed and the results commented concerning temperature effect.
Thank you for the suggestion. In the revised paper the effect of temperature is shown in Fig. 10. In the figure the temperature ranges from 27° to 40°, as shown in the recent literature [R. Rubino, S. Carrara and P. Crovetti, "Direct Digital Sensing Potentiostat targeting Body-Dust," 2022 IEEE Biomedical Circuits and Systems Conference (BioCAS), Taipei, Taiwan, 2022, pp. 280-283, doi: 10.1109/BioCAS54905.2022.9948649.].The frequency of the circuit, and in turn, the gain, GBW, phase and THD, are only slightly dependent on the temperature.
3) The dimensions of the output stage are given but the output capacitance for which this output stage has been implemented is not provided... Please provide it, and comment on the output load effect.
A fair value of 10pF was chosen for the capacitive load to account for the parasitic effect of pads, bonding, package, and the instrumentation, or to account for the circuits connected to the output of the amplifier. If the capacitive load considerably exceeds this value, a buffer stage should be added. This information was included in the revised paper at the beginning of the section 3.
Reviewer 2 Report
This paper describes a digital-based amplifier. My comments are as follows:
1. The paper should clearly describe the difference between this paper and [11].
2. The title emphasizes “investigation on the power” but the paper has very little content on “power”. For example, the analysis and simulations on the op amp’s power consumption and its relationship to noise, bandwidth, gain, etc. I would recommend the authors revise the title to better reflect the content of the paper.
3. Why is this called a “differential” amplifier if the inputs (vi’+- in Fig. 2) are not differential? In fact, the top subplot of Fig. 2 doesn’t make sense. The y-axis shows 2 signals but there is only one curve in that subplot.
4. Your common-mode vcmp is constantly moving up and down (Fig. 2 & 3). In a normal op amp you don’t want the common-mode to be constantly moving because a moving common-mode changes the op amp’s characteristic (gain, bandwidth, noise, etc.) and reduces the input range. Please explain.
5. Please include a table of performance summary and comparison to other state-of-the-art.
6. I would recommend including the following references:
Brooks, Lane, and Hae-Seung Lee. "A 12b, 50 MS/s, fully differential zero-crossing based pipelined ADC." IEEE journal of solid-state circuits 44, no. 12 (2009): 3329-3343.
Song, Yixin, Shea Smith, Benjamin Karlinsey, Aaron R. Hawkins, and Shiuh-Hua Wood Chiang. "The digital-assisted charge amplifier: A digital-based approach to charge amplification." IEEE Transactions on Circuits and Systems I: Regular Papers 69, no. 8 (2022): 3114-3123.
Kalani, Sarthak, Tanbir Haque, Rupal Gupta, and Peter R. Kinget. "Benefits of using VCO-OTAs to construct TIAs in wideband current-mode receivers over inverter-based OTAs." IEEE Transactions on Circuits and Systems I: Regular Papers 66, no. 5 (2018): 1681-1691.
N/A
Author Response
The authors thank the reviewers and the editor for the useful comments that help to improve the paper. We have carefully considered all the reviewer’s comments and we did our best to address, we hope, all his concerns. In the revised paper, all changes are marked in yellow.
Reviewer #2:
This paper describes a digital-based amplifier. My comments are as follows:
1.) The paper should clearly describe the difference between this paper and [11].
Our manuscript is a deep investigation on several aspects of the operating principles, the sizing, the issues, and the advantages of the DDA that are not discussed in [11], that are significant for designers. We have added a sentence in the revised paper.
- The title emphasizes “investigation on the power” but the paper has very little content on “power”. For example, the analysis and simulations on the op amp’s power consumption and its relationship to noise, bandwidth, gain, etc. I would recommend the authors revise the title to better reflect the content of the paper.
We have modified the title of the paper to be more correlated to the contents of the paper and the section on the power consumption and the transient behavior was improved including more details.
- Why is this called a “differential” amplifier if the inputs (vi’+- in Fig. 2) are not differential? In fact, the top subplot of Fig. 2 doesn’t make sense. The y-axis shows 2 signals but there is only one curve in that subplot.
Indeed Fig. 2 and 3 are rather cumbersome, but meaningful and of paramount importance. Although in the caption it is stated that Fig.2 shows amplifier behavior when the differential signal vd=0 i.e. vi’+ = vi’. nevertheless, we have slightly modified the figure to be more readable.
Same considerations hold for Fig. 3, when vd>0. We have modified the figure to be more readable.
- Your common-mode vcmp is constantly moving up and down (Fig. 2 & 3). In a normal op amp you don’t want the common-mode to be constantly moving because a moving common-mode changes the op amp’s characteristic (gain, bandwidth, noise, etc.) and reduces the input range. Please explain.
The DDA is not an analog op-amp: it is based on digital gates. Its behavior is indeed based on the oscillation of the common mode vcmp, as reported in the literature (by the way of example in [24], [26], [28]).
- Please include a table of performance summary and comparison to other state-of-the-art.
Our manuscript is a deep investigation on several aspects of the operating principles, the sizing, the issues, and the advantages of the DDA that are significant for designers.
A comparison between the DDA and the inverter-based and classical opamp architectures has been added at the end of section 3.
- I would recommend including the following references:
Brooks, Lane, and Hae-Seung Lee. "A 12b, 50 MS/s, fully differential zero-crossing based pipelined ADC." IEEE journal of solid-state circuits 44, no. 12 (2009): 3329-3343.
Song, Yixin, Shea Smith, Benjamin Karlinsey, Aaron R. Hawkins, and Shiuh-Hua Wood Chiang. "The digital-assisted charge amplifier: A digital-based approach to charge amplification." IEEE Transactions on Circuits and Systems I: Regular Papers 69, no. 8 (2022): 3114-3123.
Kalani, Sarthak, Tanbir Haque, Rupal Gupta, and Peter R. Kinget. "Benefits of using VCO-OTAs to construct TIAs in wideband current-mode receivers over inverter-based OTAs." IEEE Transactions on Circuits and Systems I: Regular Papers 66, no. 5 (2018): 1681-1691.
According to the reviewer’s comment we have included the suggested references.
Round 2
Reviewer 1 Report
This second version of the paper has been drastically improved and can now be published as is.
This is just a typo in the sentence just before the conclusion : "On the other hand, they are difficult to design..." instead of "On the hand they are difficult to design..."
Author Response
The authors thank again the reviewer for the useful suggestions that help to improve the paper.
The typo of the last sentence has been fixed.
Reviewer 2 Report
Regarding my comment:
"1.) The paper should clearly describe the difference between this paper and [11].
Our manuscript is a deep investigation on several aspects of the operating principles, the sizing, the issues, and the advantages of the DDA that are not discussed in [11], that are significant for designers. We have added a sentence in the revised paper."
Please include descriptions of the differences IN THE BODY. Also, the descriptions should be clear and specific so readers understand the differences from [11]. The current descriptions are too vague. You should summarize what was discussed in [11], then describe how this paper differs from [11] in detail.
Author Response
"The paper should clearly describe the difference between this paper and [11]...."
the original ref. [11] is now ref. [24]. Our paper is a deep investigation on the schematic presented in [24] and in the related literature. No different topologies are proposed in our manuscript. Moreover, the differences between our work and the one in [24] are already explained in the abstract: investigation on power consumption and operating regions, and sizing of the circuit in an integrated technology (UMC 180nm).